# Novel IMB16-4 Compound Loaded into Silica Nanoparticles Exhibits Enhanced Oral Bioavailability and Increased Anti-Liver Fibrosis In Vitro

**DOI:** 10.3390/molecules26061545

**Published:** 2021-03-11

**Authors:** Xia Niu, Xiaomei Wang, Bingyu Niu, Guoqing Li, Xinyi Yang, Yucheng Wang, Guiling Li

**Affiliations:** Institute of Medicinal Biotechnology, Chinese Academy of Medical Science & Peking Union Medical College, Beijing 100050, China; niuxia307@163.com (X.N.); w18810685672@126.com (X.W.); niubingyu@imb.pumc.edu.cn (B.N.); gli@imb.cams.cn (G.L.); yangxinyi1976@hotmail.com (X.Y.)

**Keywords:** mesoporous silica nanoparticle, liver fibrosis, oral bioavailability, TGF-β1, dissolution rate

## Abstract

Background: Liver fibrosis, as a common and refractory disease, is challenging to treat due to the lack of effective agents worldwide. Recently, we have developed a novel compound, N-(3,4,5-trichlorophenyl)-2(3-nitrobenzenesulfonamide) benzamide (IMB16-4), which is expected to have good potential effects against liver fibrosis. However, IMB16-4 is water-insoluble and has very low bioavailability. Methods: Mesoporous silica nanoparticles (MSNs) were selected as drug carriers for the purpose of increasing the dissolution of IMB16-4, as well as improving its oral bioavailability and inhibiting liver fibrosis. The physical states of IMB16-4 and IMB16-4-MSNs were investigated using nitrogen adsorption, thermogravimetric analysis (TGA), HPLC, UV-Vis, X-ray diffraction (XRD) and differential scanning calorimetry (DSC). Results: The results show that MSNs enhanced the dissolution rate of IMB16-4 significantly. IMB16-4-MSNs reduced cytotoxicity at high concentrations of IMB16-4 on human hepatic stellate cells LX-2 cells and improved oral bioavailability up to 530% compared with raw IMB16-4 on Sprague–Dawley (SD) rats. In addition, IMB16-4-MSNs repressed hepatic fibrogenesis by decreasing the expression of hepatic fibrogenic markers, including α-smooth muscle actin (α-SMA), transforming growth factor-beta (TGF-β1) and matrix metalloproteinase-2 (MMP2) in LX-2 cells. Conclusions: These results provided powerful information on the use of IMB16-4-MSNs for the treatment of liver fibrosis in the future.

## 1. Introduction

Liver fibrosis is a global health problem and is characterized as cholestasis, hepatitis and nonalcoholic fatty liver disease [1,2,3,4]. During hepatic injury, a quiescent state of hepatic stellate cells (HSCs) in healthy liver is activated. Tissue inhibitors of matrix metalloproteinases (TIMPs) are synthesized and secreted by activated HSCs [5,6], further preventing degradation of the extracellular matrix (ECM) by matrix metalloproteinases (MMPs) [7,8,9]. The accumulation of ECM results in liver fibrosis. Numerous studies have elucidated the various factors and cell signaling pathways that regulate HSC activation, including transforming growth factor-beta (TGF-β) [10]. TGF-β is the strongest known inducer of hepatic fibrogenesis, which activates the transcription of the key gene collagen type Iα1chain (COL1A1) and upregulates protein expression, especially collagen type I (COL1) and α-smooth muscle actin (α-SMA) [11,12,13].

Therefore, approved treatments for liver fibrosis are very poor [14,15,16,17] and there is an urgent need to develop novel agents to treat liver fibrosis. Recently, we synthesized a series of novel compounds, such as N-(3,4-dichlorophenyl)-2-(3-trifluoromethoxy) benzamide, which is also referred to as IMB17-15. IMB17-15 has hepatoprotective effects on nonalcoholic fatty liver disease (NAFLD) model rats [18,19]. Among them, N-(3,4,5-trichlorophenyl)-2(3-nitrobenzenesulfonamide) benzamide, abbreviated as IMB16-4, the sister compound of IMB17-15, was synthesized. In this study, we made an effort to explore the pharmaceutical properties of IMB16-4 and its effect on resisting liver fibrosis. However, IMB16-4 displays very poor aqueous solubility and poor oral bioavailability. To overcome this hurdle, an oral delivery vehicle for IMB16-4 is of great necessity to improve its pharmacokinetic properties [20].

Mesoporous silica nanoparticles (MSNs) have attracted great attention because of their special features such as their unique porous structure, large surface area, pore volume and strong absorbability [21,22,23,24,25]. All these features allow better control of drug loading and release. Poorly water-soluble drug molecules are loaded into small mesopores and exist in noncrystalline form by decreasing the Gibbs free energy of the system [26]. This noncrystalline form can be rapidly released from mesopores and generate supersaturated solutions on a silica surface [27]. This ability of silica nanoparticles to increase the drug dissolution rate plays an important role in enhancing oral bioavailability [28]. In addition, numerous factors including particle size, pore diameter, pore length and modification of the surface could affect drug release from MSMs, leading to drug release occurring within a few minutes or within days [29]. Furthermore, MSNs can protect drugs from hydrolysis, oxidation, or degradation processes due to the physicochemical stability of the silica matrix [30].

In this paper, we report for the first time that MSNs were used as the carrier for IMB16-4 to increase the dissolution rate, improve bioavailability and inhibit liver fibrosis effects. MSNs with relatively large pore diameter and short pore channel length were applied to improve the dissolution rate and achieve controlled release. The effects of MSNs on the uptake and release of IMB16-4 were systematically studied using SEM, TEM, N_2_ adsorption, XRD, differential scanning calorimetry (DSC), TGA and HPLC. In vivo pharmacokinetic studies were conducted to confirm the enhancement of oral bioavailability. In addition, the anti-fibrotic effects of IMB16-4 loaded into MSNs was also explored in vitro on the human HSC line LX-2 cells by testing TGF-β1, α-SMA and MMP2 protein activity.

## 2. Results and Discussion

### 2.1. Morphology of MSNs and IMB16-4 MSNs

The morphology and particle size of MSNs and IMB16-4-MSNs were analyzed by SEM and TEM. As shown in Figure 1A,B, MSNs had a nearly monodispersed spherical shape with a size of about 60 nm. The mean pore sizes of MSNs were approximately 8 nm and were both on the particle surface and within the particle. As seen in Figure 1C,D, the pores of MSNs were partly blocked by IMB16-4 (Figure 1C). Obviously, the channels of pores were seriously blocked when the mass ratio of IMB16-4 and MSNs was 1:1 (Figure 1D).

### 2.2. Estimation of the Brunauer–Emmett–Teller (BET) Specific Surface Area from Nitrogen Adsorption Studies

The nitrogen adsorption/desorption isotherms of samples are presented in Figure 2. The values for the BET specific surface area (S_BET_), the total pore volume (Vt) and the Barrett–Joyner–Halenda (BJH) pore diameter (w_BJH_) are given in Table 1. MSNs possess high S_BET_, Vt and large pore diameter, which indicate its ability to store small agents [21]. After being loaded with IMB16-4, S_BET_, Vt and w_BJH_ were drastically reduced. This was due to the fact that IMB16-4 was loaded into the pores.

### 2.3. Quantification of IMB16-4 Uptake by TGA and HPLC Analysis

In order to achieve the maximum drug loading, MSNs were soaked in DMSO solution during drug loading [31]. Then, DMSO was removed by vacuum drying at 80 °C and 178 °C, successively. Afterwards, the amount of drug loading was quantified by TGA and HPLC, respectively. In the TGA measurement, drug loading was obtained by a temperature-dependent weight reduction. MSNs showed good thermostability and the weight reduction was mainly attributed to the IMB16-4. The weight lossresidual rates of raw IMB16-4, IMB16-4-MSNs (1:1) and IMB16-4-MSNs (1:2) were 62.7%, 32.8% and25.8%, respectively (Figure 3A). The drug loading was 41.1% and 52.3% for IMB16-4-MSNs (1:2) and IMB16-4-MSNs (1:1), respectively, which was calculated by the ratio of the weight loss of IMB16-4-MSNs to the weight loss of raw IMB16-4. It is noted that TGA measurement is not sufficiently accurate to quantify the total drug content of the sample. Nevertheless, together with HPLC, TGA is an important method for detecting drug loading. As shown in Figure 3B, a strong absorption peak at 258 nm was observed, which was attributed to IMB16-4. MSNs did not affect absorption at 258 nm. Therefore, 258 nm was selected as the detection wavelength of drug loading. The drug uptake values obtained by HPLC are shown in Table 1.

### 2.4. Solid State Characterization Using DSC and XRD Studies

The crystalline form can be estimated by DSC analysis when melting point depression appears. If the compound is present in a noncrystalline state, no melting point depression can be detected [32]. As shown in Figure 4, the DSC curve of IMB16-4 exhibited a single endothermic peak at 253 °C ≈ 258 °C, which corresponded to its intrinsic melting point. The melting point depression of the physical mixture also appeared. However, no melting peak was observed in the IMB16-4-MSNs (the mass ratio of IMB16-4 and MSNs was 1:2.), indicating the absence of a noncrystalline state. These assumptions could be further confirmed by the results of XRD study.

The XRD patterns of the IMB16-4-MSNs samples were recorded to determine whether a crystalline IMB16-4 phase could be detected. As shown in Figure 5, the diffraction pattern of raw IMB16-4 was highly crystalline, as indicated by the numerous peaks. For the physical mixture, peaks were attributed to samples of pure IMB16-4. However, the diffraction profile of IMB16-4-MSNs (the mass ratio of IMB16-4 and MSNs was 1:2) showed no peaks. It is known that the absence of distinctive peaks indicates that the IMB16-4 loaded into the pores of MSNs exists in a noncrystalline state. In contrast, slight peaks of IMB16-4-MSNs (the mass ratio of IMB16-4 and MSNs was 1:1) were observed, indicating that MSNs reached the limit of suppressing crystallization.

### 2.5. Effects of MSNs on IMB16-4 Release Behavior

As shown in Figure 6, the observed dissolution rate of the raw IMB16-4 was quite low and the amount of dissolved IMB16-4 in the release medium was about 40% at 12 h. However, the amounts of dissolved IMB16-4 from IMB16-4-MSNs (1:2) in pH6.8 phosphate buffer solutions at sampling times of 1 h, 6 h and 12 h accumulated to 32.4%, 56.3% and 66.8%, respectively. The corresponding amounts were 16.9%, 35.1% and 44.7% for IMB16-4-MSNs (1:1). Remarkably, the dissolution rate of IMB16-4 released from IMB16-4-MSNs was faster compared with that of raw IMB16-4, especially as the mass ratio of IMB16-4 and MSNs was 1:2. This dissolution improvement may be mainly attributed to the large pores of MSNs maintaining nanoscale IMB16-4 and transforming the crystalline state of IMB16-4 to a noncrystalline state, which is known to improve the drug dissolution rate. It can also be seen that the release profiles of all IMB16-4-MSN samples were of the same unique type with a sustained release of IMB16-4 both in pH6.8 and pH 1.0 release medium. The initial bursts of IMB16-4 release were attributed to the presence of IMB16-4 in the external pores and near the holes of the MSNs, which allows a certain amount of IMB16-4 to be released quickly into the release medium and satisfies the need for immediate treatment after administration. The release rate then became slower, due to the slow dissolution of IMB16-4 from the pores inside the particles. The diffusion of solvent into the small mesopores and the counter diffusion of IMB16-4 out of the mesopore channels delayed the release of IMB16-4.

### 2.6. The Cytotoxicity on Human Hepatic Stellate Cells (LX-2 Cells)

Figure 7 shows the survival rate of LX-2 cells after 24 h co-culture. IMB16-4-MSNs at the concentration of 1~100 μM exhibited no cytotoxicity to LX-2 cells. The tested concentration of MSNs was equivalent to the content of MSNs in corresponding IMB16-4-MSNs and showed no cytotoxicity to LX-2 cells. At the same time, when the concentration of IMB16-4-MSNs and raw IMB16-4 increased, the cell survival rate was reduced, and this toxic effect observed at high concentrations was absolutely attributed to raw IMB16-4. In addition, the cell survival rate of IMB16-4-MSNs at 100 μM was higher compared with that of raw IMB16-4. The results indicated that MSNs reduced the cytotoxicity of IMB16-4-MSNs.

### 2.7. In Vitro Antifibrotic Effects

LX-2 cells were stimulated with TGFβ1 protein (2 ng/mL) and then treated with raw IMB16-4 and IMB16-4-MSNs, respectively. In vitro antifibrotic effects were evaluated by the expression of hepatic fibrogenic markers, including α-SMA, TGF-β1 and MMP2 using Western blot. After stimulation with TGFβ1, the expression of hepatic fibrogenic markers was increased. Along with the IMB16-4 treatment, the regulated expression of all hepatic fibrogenic markers is shown in Figure 8. IMB16-4-MSNs significantly decreased the protein levels of α-SMA, TGF-β1 and MMP2 on LX-2 cells. The antifibrotic effects of IMB16-4-MSNs were stronger than those of raw IMB16-4, indicating that IMB16-4-MSNs inhibited liver fibrosis effects. After being dissolved by DMSO, IMB16-4 showed potential anti-liver fibrosis effects. However, the differences are not significant. It can be assumed that IMB16-4 in DMSO was separated out in a serum-free culture. IMB16-4, in sterile water, possessed a poor inhibition effect, attributed to poor solubility (43.2 ± 9.1 ng/mL) and far from the concentration required to show anti-liver fibrosis effects. Overall, IMB16-4-MSNs increased inhibition of liver fibrosis effects, owing to the MSNs reducing the crystallite size, increasing dispersibility and enhancing solubility.

### 2.8. MSNs Increase IMB16-4 Absorption In Vivo

The in vivo bioavailability of IMB16-4 from IMB16-4-MSNs (1:2) was assessed with raw IMB16-4 as a control. As shown in Figure 9 and Table 2, the Cmax and AUC_0≈12 h_ values of IMB16-4-MSNs (1:2) after intragastric administration were increased nearly 5-fold compared with raw IMB16-4. The plasma concentrations of IMB16-4 were greatly promoted by the MSNs. The Cmax values for IMB16-4-MSNs and raw IMB16-4 were 0.89 ± 0.22 and 0.18 ± 0.04 mg/L, respectively. The AUC_0≈12 h_ value for IMB16-4-MSNs was increased about 5.3-fold compared with that of raw IMB16-4. Therefore, clearly, IMB16-4-MSNs significantly improved the in vivo adsorption of IMB16-4. The mean Tmax values for IMB16-4-MSNs and raw IMB16-4 were 4.14 h and 3.43, respectively. The large pore diameter, short pore channel, protective effect and nondegradable nature of MSNs may make the IMB16-4 release behavior as effective in vivo as that in vitro. In addition, several factors attributed to the improvement of oral bioavailability, including greater dissolution rate of IMB16-4 owing to its noncrystalline state, reduced crystallite size and increased dispersibility.

## 3. Materials and Methods

### 3.1. Materials

Tetraethyl orthosilicate (TEOS) and hexadecyltrimethylammonium bromide (CTAB) were purchased from Aladdin (Shanghai, China). 2,2′-Azobis (2-methylpropionamide) dihydrochloride (AIBA, ≥99%) was purchased from Shanghai Yuanye Bio-Technology Co., Ltd. L-lysine was purchased from Rhawn reagent. Recombinant human TGF-β1 protein (TGF-β1) was obtained from R&D Systems (R&D, USA). Antibodies for MMP2 and α-SMA were obtained from Abcam (Abcam, UK). TGF-β1 polyclonal antibody was purchased from Abnova (Abnova, USA). Antibody for glyceraldehyde-3-antiphosphate dehydrogenase (GAPDH) and horseradish peroxidase (HRP)-conjugated secondary antibodies against mouse or rabbit IgG were obtained from Proteintech (Wuhan, China). All the other reagents were of analytical grade and were obtained from commercial sources.

Sprague–Dawley (SD) rats were supplied by HFK Biotechnology Co. Ltd. (Beijing, China). All animal experiments were approved by the local IACUC (Institutional Animal Care and Use Committee, Beijing, China) and performed in accordance with the Institutional Review Board for Laboratory Animal Care (IMB-2020121406D6).

### 3.2. Preparation of MSNs

MSNs were synthesized as reported [33,34]. In total, 600 mg of CTAB was dissolved in 192 mL of deionized water at 60 °C under vigorous stirring in a three-necked flask reactor for 30 min. Then, 152 mL of octane, 132 mg of L-lysine, 289 mg of AIBA, 6.4 mL of TEOS and 21 mL of styrene monomer were added to the system in sequence. The reaction was kept for 4 h under nitrogen at 60 °C with constant stirring. Afterwards, the resulting product was cooled to room temperature over one night and purified by centrifugation at a rate of 15,000 rpm. Then the precipitate was washed with ethanol. After centrifugation, the precipitate was heated at 600 °C for 3 h under atmospheric conditions to remove organic template.

### 3.3. Drug Loading by Solvent Evaporation Method

*N*-(3,4,5-*trichlorophenyl*)-2(3*-nitrobenzenesulfonamido*) *benzamide* (**IMB16-4**). White solid, m.p.: 256.3 ≈ 258.1 °C. ^1^H-NMR δ 10.55 (s, 1H), 10.34 (s, 1H), 8.44 (s, 1H), 8.35 (dd, *J* = 8.5, 2.1 Hz, 1H), 8.10 (d, *J* = 7.7 Hz, 1H), 7.88 (s, 2H), 7.78 (t, *J* = 8.0 Hz, 1H), 7.63 (d, *J* = 7.6 Hz, 1H), 7.56 (t, *J* = 7.7 Hz, 1H), 7.43 ≈ 7.32 (m, 2H). MS m/z: Calcd for C_19_H_12_Cl_3_O_5_S 499.2 [M−H]^−^.

In total, 10 mL DMSO solution of IMB16-4 (30 mg/mL) was dropped to 300 and 600 mg MSNs, separately. The corresponding mass ratio of IMB16-4 and MSNs was 1:1 and 1:2. The mixture was gently stirred and then DMSO was evaporated by a rotary evaporator at 80 °C. The residual DMSO was removed at 178 °C under the vacuum. The resulting samples were referred to as IMB16-4-MSNs.

### 3.4. Equilibrium Concentration Study

An excess amount of raw IMB16-4 was added into 5 mL of distilled water at 25 ± 2 °C. The raw IMB16-4 solution at equilibrium time (about 24 h) was withdrawn and filtered using a 0.22 μm membrane. The subsequent filtrate was diluted with internal standard (4′-Chloroacetanilide) acetonitrile solution and analyzed by HPLC-MS/MS analysis (Thermo LTQ XL, USA). The standard curve for IMB16-4 was the linear (R^2^ > 0.992) over the concentration range of 0.3~1200.0 ng/mL. The quantitative ion pair of IMB16-4 and internal standard quantitative were m/z = 499.9/196.0 and m/z = 168.04/126.1, respectively.

### 3.5. Sample Characterization

The porous structure, morphology and particle size of MSNs and IMB16-4-MSNs were evaluated using a TEM (JEM1200EX, JEOL, Japan) and an SEM (SU8020, HITACHI, Japan). A very low concentration of samples was dispersed in water under ultrasonication, then dropped onto gold-plated and carbon-coated copper grids, respectively. The pore characteristics of the samples were studied by determining the nitrogen adsorption using a surface area and pore size analyzer (ASAP 2460, micromeritics, USA). The samples were outgassed at 150 °C for 6 h prior to analysis. The pore characteristics were determined according to the BET and BJH procedures from the desorption branches of the isotherms. The physical state of IMB16-4 was evaluated using an X-ray diffractometer (Brucker D8 Advance, Germany). Data were obtained from 5° to 40° (diffraction angle 2θ) at a step size of 0.02° and a scanning speed of 4°/min radiation. DSC analysis of the samples was examined by differential scanning calorimetry (DSC 1, Mettler, Switzerland). The samples were heated over a temperature range between 50 and 300 °C at a rate of 10 °C/min under a nitrogen purge of 40 mL/min. The drug loading was examined by TGA (TGA/DSC 1, Mettler, and Switzerland). The samples were heated over a temperature range between 40 and 900 °C at a rate of 10 °C/min under a nitrogen purge of 50 mL/min.

### 3.6. HPLC Analysis

The samples were analyzed by HPLC (Nexera-i LC-2040C 3D, Shimadzu, Japan). Analysis was carried out on a Shim-pack GIST C18 column (50 × 2.1 mm, 2 μm, Shimadzu). The mobile phase consisted of an 80:20 (% *v*/*v*) mixture of methanol and pH 2.0 phosphoric acid solutions, the flow rate was 0.3 mL/min and the detection wavelength was 258 nm. The column temperature was 25 °C. The injection volume was 5 μL. The retention times are about 3.07 min for IMB16-4. The samples were filtered using a 0.22 μm membrane filter before running the HPLC analysis. Drug loading (%) = (weight of IMB16-4 in samples/weight of samples) × 100.

### 3.7. In Vitro Dissolution

Dissolution studies were conducted using a USP II paddle method (100 rpm, 37 °C, and 900 mL dissolution medium) with a dissolution tester (ZRS-8LD, China). The release of IMB16-4 and IMB16-4-MSNs was performed in pH 1.0 hydrochloric acid solution and pH 6.8 phosphate buffer solutions, containing 3% (*w*/*v*) SDS, respectively. All release studies were carried out in triplicate. At predetermined time intervals, 5 mL of sample solution was withdrawn from the release medium and filtered using a 0.22 μm membrane filter before running the HPLC analysis. An equivalent amount of fresh medium was added to maintain a constant dissolution volume.

### 3.8. In Vitro Cytotoxicity

The human HSC line LX-2 was obtained from Pro He [35]. LX-2 cells were cultured in dulbecco’s modified eagle medium (DMEM)/GlutaMAX I (Invitrogen, USA) with 10% fetal bovine serum and 1% penicillin/streptomycin at 37 °C in an atmosphere of 5% CO_2_. The cell suspension was seeded into 96-well plates at 100 μL per well and incubated for 24 h. Then, MSNs, IMB16-4 and IMB16-4-MSNs suspensions containing different concentrations were added to 96-well plates at 100 μL per well and incubated for 24 h. Then, 10 μL CCK8 solution was added to 96-well plates and incubated for 2 h. Finally, the absorbance was determined at 450 nm by an optical microscope (BioTek, SYNERGYH1, USA). The cell survival rate was calculated according to formula: Cell survival rate (%) = Absorbance of sample/Absorbance of control×100. 

### 3.9. The Antifibrotic Effects on the Human HSC Line LX-2 Cells

After ultraviolet sterilization, raw IMB16-4, IMB16-4-MSNs and MSNs were suspended in water, respectively. Another group of raw IMB16-4 was suspended in DMSO. The concentration of IMB16-4 was 2 mM. Then, 2µL of solution containing IMB16-4 was added into 2 mL of serum-free culture with TGFβ1. LX-2 cells were cultured as described above. LX-2 cells were seeded in a 6-well plate, cultured in DMEM/GlutaMAX I, containing 10% fetal bovine serum (FBS) in 5% CO_2_ atmosphere at 37 °C. Serum-free culture was replaced until the cells reached 90 ≈ 95% confluence. After 24 h, cells were treated with TGFβ1 (2 ng/mL) and IMB16-4 (2 µM) for 24 h. Then, cells were washed with phosphate buffer saline (PBS) and protein was extracted in radio-immunoprecipitation assay (RIPA) buffer. Then mixture was centrifuged at 12,000 rpm at 4 °C for 20 min. The supernatant was collected, and the total protein was determined using the bicinchoninic acid (BCA) protein assay kit (Beyotime Biotechnology, China). Samples were applied to the 10% SDS-PAGE gel. Then, the protein bands were transferred to a polyvinylidene difluoride (PVDF) membrane (Millipore Corp, Atlanta, GA, USA) and the membrane was blocked for 1 h with 5% nonfat milk. The membrane was then incubated with desired primary antibodies overnight at 4 °C followed by horseradish peroxidase (HRP) conjugated secondary antibodies (1:10,000) at room temperature for 1 h. The protein bands were analyzed on an imager (Tanon5200, China).

### 3.10. Pharmacokinetics Study

The mice were maintained under a specific pathogen-free (SPF) environment with a 12 h light/dark cycle. SD rats (body weight 200 ± 20 g) were fasted overnight and divided into two groups. Raw IMB16-4 and IMB16-4-MSNs were given orally by gavages at a dose of 100 mg/kg, respectively. Before administering, both IMB16-4-MSNs and raw IMB16-4 were dispersed in 0.5% CMC-Na aqueous solution, respectively. Blood samples were collected from the eye socket vein at time points of 0.17, 0.5, 1, 2, 3, 4, 6, 8 and 12 h after dosing. Plasma samples were collected by centrifuging at 3000 rpm for 10 min and were then stored at −20 °C until analysis. Plasma samples were mixed with internal standard (4′-Chloroacetanilide) acetonitrile solution. Then, the mixture was vortex-mixed for 10 s. After centrifugation at 15,000 rpm for 10 min, the supernatant was used for HPLC-MS/MS analysis (AB SCIEX 6500 Qtrap, USA). The condition of the capillary voltage was −4500 V. The temperature was 500 °C. The mobile phase comprised acetonitrile and 10 mM of ammonium acetate aqueous solution. Analysis was carried out on an Xselect HSS T3 column (2.1 × 100 mm, 2.5 μm, Waters) in gradient elution. The column temperature was 40 °C and the flow rate was 0.3 mL/min. The standard curve for IMB16-4 was the linear (R^2^ > 0.992) over the concentration range of 4~2048 ng/mL. The quantitative ion pairs of IMB16-4 and internal standard quantitative were m/z = 500.0/195.7 and m/z = 168.0/126.1, respectively. The pharmacokinetic parameters were obtained using statistic software DAS2.0.

### 3.11. Statistical Analysis

Each experiment was performed at least in triplicate. Statistical analysis was performed by one-way or two-way analysis of variance (ANOVA) followed by Dunnett’s multiple comparison tests using SPSS 13.0 and Origin 9.1 software. Statistical significance was accepted at the level of *p* < 0.05.

## 4. Conclusions

IMB16-4, as a novel compound with potential anti-liver fibrosis effects, was loaded into MSNs to enhance oral bioavailability and improve therapeutic efficacy by changing the crystalline state of IMB16-4, regulating release from silica nanocavities. The advantages of MSNs with large pore diameter and short pore channel were major contributing factors for release. The dissolution of IMB16-4-MSNs in the release medium showed great advantages compared with that of raw IMB16-4, which accounted for the enhanced oral bioavailability and inhibited liver fibrosis effect. Overall, these results offer powerful information presenting IMB16-4-MSN as an ideal anti-liver fibrosis preparation.

## Figures and Tables

**Figure 1 molecules-26-01545-f001:**
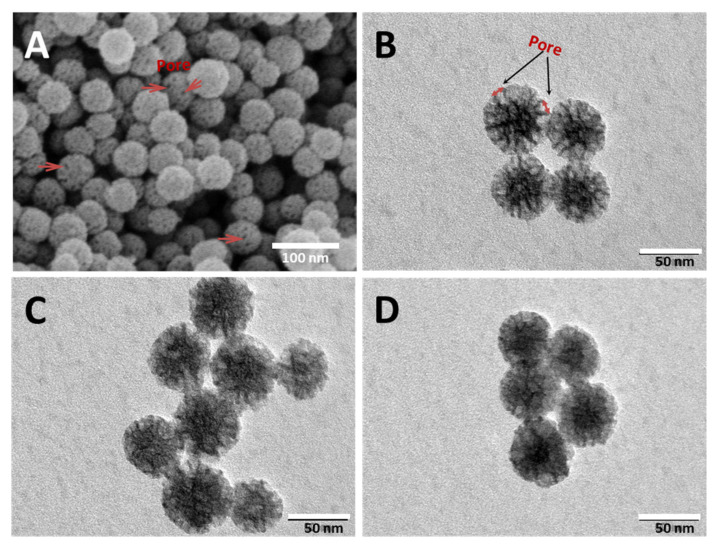
SEM photographs of mesoporous silica nanoparticles (MSNs) (**A**) and TEM photographs of MSNs (**B**), IMB16-4-MSNs 1:2 (**C**) and IMB16-4-MSNs 1:1 (**D**).

**Figure 2 molecules-26-01545-f002:**
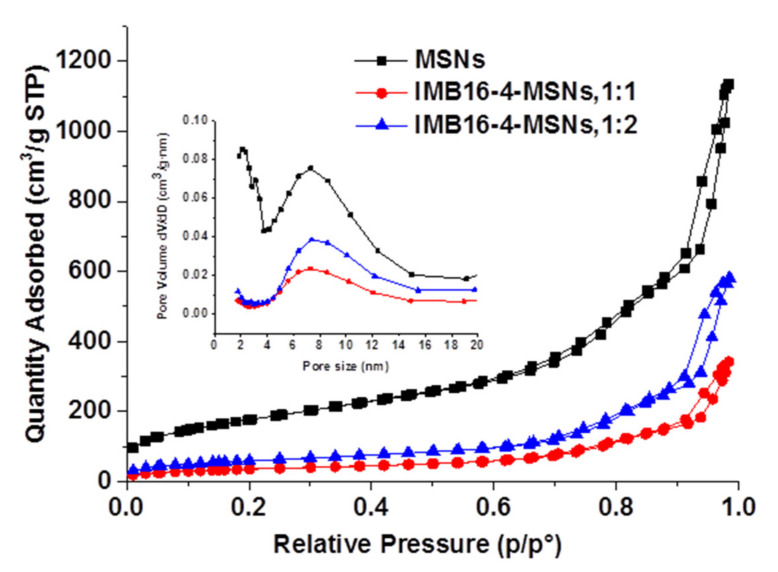
N_2_ adsorption–desorption isotherms and the pore size distribution of MSNs and loaded samples (inner).

**Figure 3 molecules-26-01545-f003:**
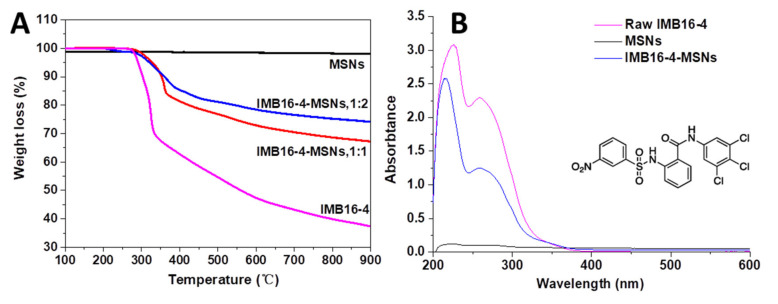
TGA of MSNs and IMB16-4-MSNs samples (**A**). UV-Vis absorption spectra of raw IMB16-4, MSNs, IMB16-4-MSNs and chemical structural of IMB16-4 (**B**).

**Figure 4 molecules-26-01545-f004:**
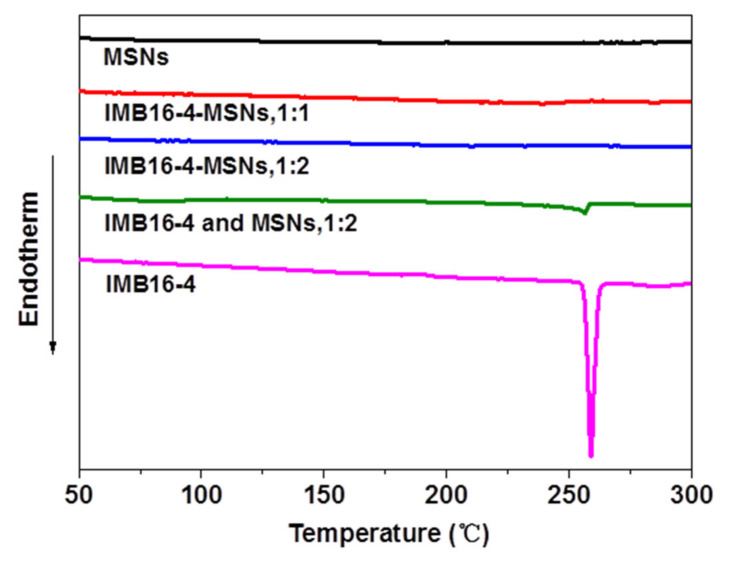
Differential scanning calorimetry (DSC) graphs of MSNs, physical mixture and IMB16-4-MSNs.

**Figure 5 molecules-26-01545-f005:**
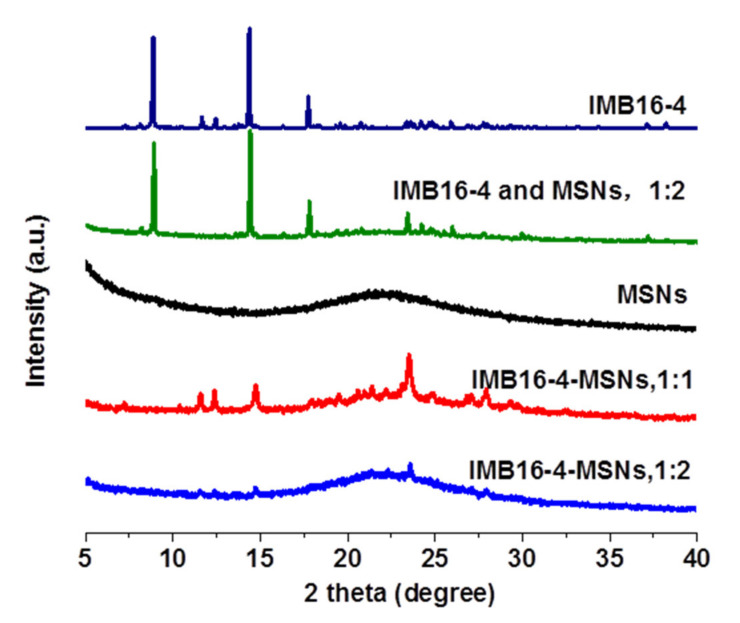
XRD patterns of MSNs, physical mixture and IMB16-4-MSNs.

**Figure 6 molecules-26-01545-f006:**
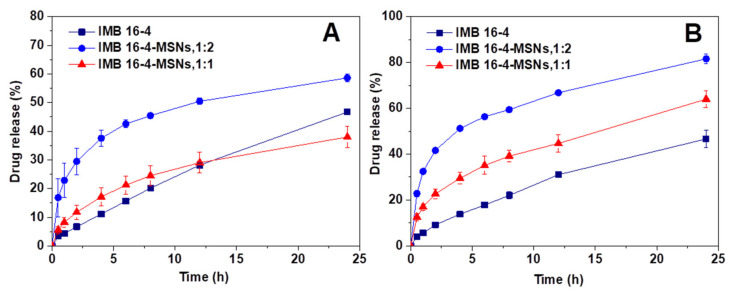
Dissolution profiles of IMB16-4 and loaded samples in pH 1.0 hydrochloric acid containing 3% SDS (**A**) and pH 6.8 phosphate buffer solutions containing 3% SDS (**B**). Each data point represents the mean ± SD of three determinations.

**Figure 7 molecules-26-01545-f007:**
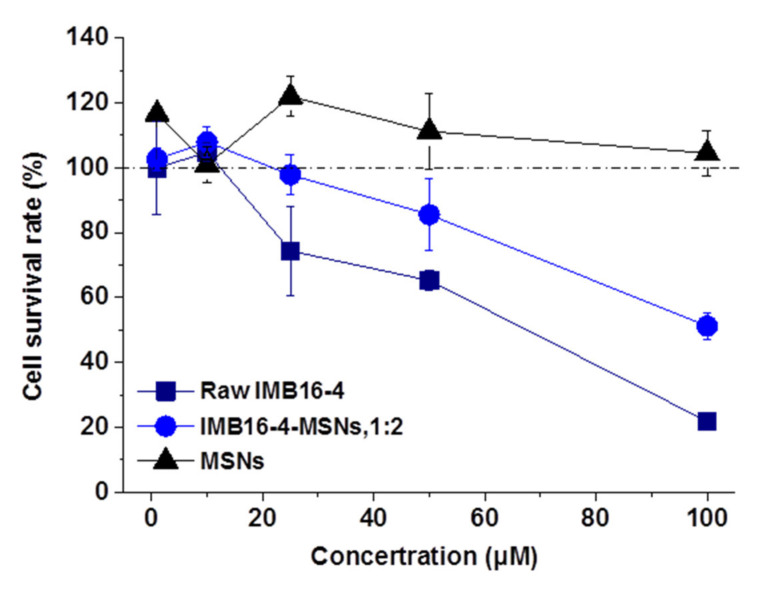
The cell survival rate of LX-2 cells at varying concentrations of raw IMB16-4-MSNs and IMB16-4-MSNs (1:2). Each data point represents the mean ± SD of three determinations.

**Figure 8 molecules-26-01545-f008:**
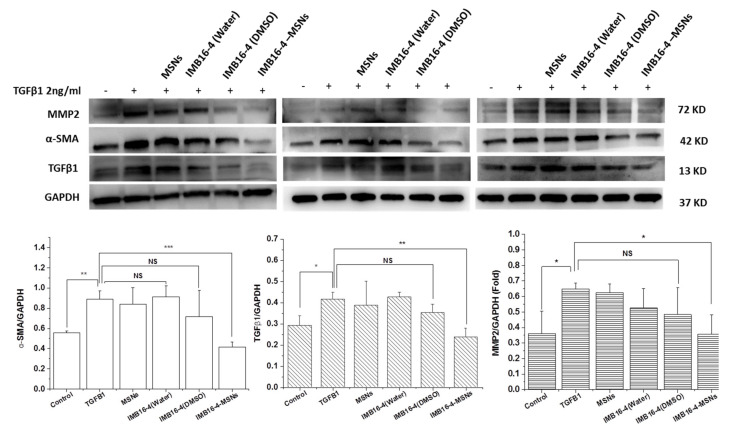
Effects of IMB16-4 and IMB16-4-MSNs on inhibiting fibrogenetic α-SMA, TGF-β and MMP2 protein levels in LX-2 cells. Proteins were extracted and analyzed by Western blot in the experimental section. The values are expressed as the mean ± SD of triplicate independent experiments (*** *p* < 0.001, ** *p* < 0.01 and * *p* < 0.05 vs. the TGFβ1 group).

**Figure 9 molecules-26-01545-f009:**
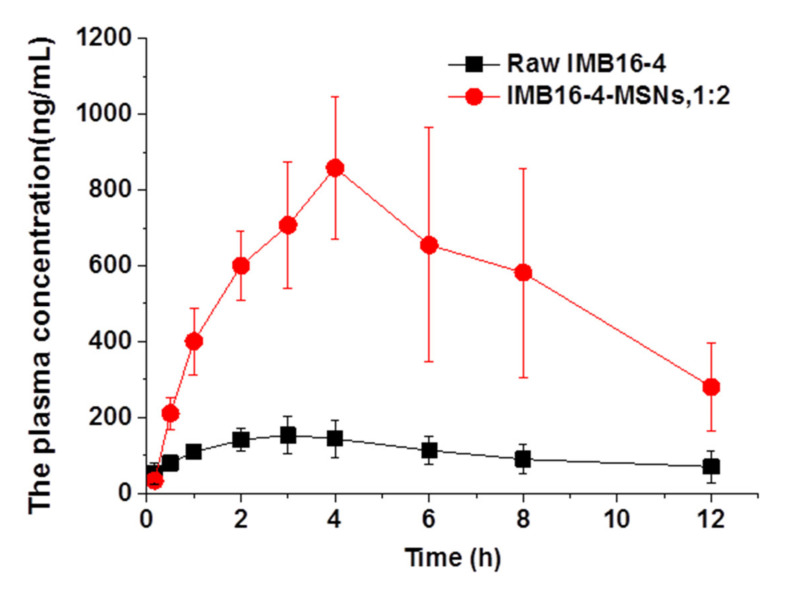
Plasma concentration–time profiles of raw IMB16-4, MSNs and IMB16-4-MSNs (1:2). Results are expressed as the mean with the bar showing SD values of seven rats.

**Table 1 molecules-26-01545-t001:** Properties characterization for MSNs and IMB16-4-MSNs. Each data point represents the mean ± SD of three determinations.

Sample	S_BET_ (m^2^/g)	Vt (cm^3^/g)	*w*_BJH_ (nm)	Drug Loading (% TGA)	Drug Loading (% HPLC)
MSNs	636.7	1.73	7.1	—	—
IMB16-4-MSNs, 1:1	145.2	0.52	7.2	52.3	44.4 ± 2.25
IMB16-4-MSNs, 1:2	241.2	0.89	7.2	41.1	30.7 ± 0.61

**Table 2 molecules-26-01545-t002:** Pharmacokinetic parameters of raw IMB16-4, MSNs and IMB16-4-MSNs (1:2), mean ± SD, *n* = 7.

Pharmacokinetic Parameter	Raw IMB16-4	IMB16-4-MSNs (1:2)
C_max_ (mg/L)	0.18 ± 0.04	0.89 ± 0.22
T_max_ (h)	3.43 ± 1.40	4.14 ± 0.90
AUC_0≈12 h_ (mg/L h)	1.24 ± 0.31	6.61 ± 1.72
MRT_0≈12 h_	5.18 ± 0.77	5.64 ± 0.29
CLz/F	54.70 ± 23.57	11.98 ± 3.56
F	100%	533%

## Data Availability

Not applicable.

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
