# Peer review of "Novel IMB16-4 Compound Loaded into Silica Nanoparticles Exhibits Enhanced Oral Bioavailability and Increased Anti-Liver Fibrosis In Vitro"

_molecules, 2021, doi:10.3390/molecules26061545_

Round 1

Reviewer 1 Report

Few sentences has to be rephrased in the introduction.

  1. 'So for, approved treatment for liver fibrosis is very poor. Many potential antifibrotic compounds are being tested in clinical studies '.
  2. 'However, as innovative agent, IMB16-4 displays very poor aqueous solubility, resulting in poor oral bioavailability due to insufficient dissolution throughout the gastrointestinal tract. It is an obstacle on the drug discovery programs. To overcome this hurdle, oral delivery vehicle of IMB16-4 is of great necessity to improve the pharmacokinetic properties [20]. '

Representation of degree centigrade, pH is not uniform through the text.

Author Response

I quite appreciate your favorite consideration. 

Response 1: I am sorry making an ambiguous statement. " Many potential antifibrotic compounds are being tested in clinical studies " was deleted.

Response 2: I will revise it as follows"However, IMB16-4 displays very poor aqueous solubility and poor oral bioavailability.To overcome this hurdle, oral delivery vehicle of IMB16-4 is of great necessity to improve the pharmacokinetic properties .  "

Representation of degree centigrade and pH has been revised and is uniform through the text.

Reviewer 2 Report

Dear authors,

We recommend small revisions for text editing, as follows:

  • all "in vivo" and "in vitro" entries should be written with italics;
  • S.D., or S.D, or SD (collation);
  • sometimes, using "anti-" in the entire manuscript creates an anachronic state, as the treatment is "against" or "for" hepatic fibrosis, and not "anti-hepatic fibrosis," canceling/nullifying the meaning itself;
  • within Abstract (suggestions):

- "because of the lack" ➦ "due to the lack" (in this case, a lack keeps the negative meaning, and does not enhance a fact);

- "to have good potential of anti-liver fibrosis" ➦ "to have good potential against liver fibrosis";

- "to increase the dissolution, improve the oral bioavailability and enhance the anti-liver fibrosis of IMB16-4" ➦ "For increasing the dissolution, improving the oral bioavailability and enhancing the IMB16-4 effect against liver fibrosis";

- "These results provided powerful information for IMB16-4-MSNs as anti-liver fibrosis treatment in the furture" ➦ "These results provided powerful information on IMB16-4-MSNs for the treatment of liver fibrosis in the future";

  • within Introduction (suggestions):

- "So for" ➦ "Therefore";

-  "In this paper, we report for the first time the application of MSNs as the vehicle of IMB16-4 increased dissolution rate, improved bioavailability and anti-liver fibrosis effect" ➦ might nanoparticles be vehicles of a dissolution rate, bioavailability and effect? Please revise!!!

  • Subchapter 2.3 (suggestions):

- "MSNs was soaked" ➦ "MSNs were soaked";

- "The residua rate… were" ➦ "The residual rates… were";

  • Subchapter 2.4 (suggestions):

- "These assume could be further confirmed by the results" ➦ "These assumptions could be further confirmed by the results";

  • Subchapter 2.5 (suggestions):

- "may be mainly attributed to the large mesopores of MSNs" ➦ "may be mainly attributed to the large pores of MSNs";

  • Subchapter 2.6 (suggestions):

- "Fig.7 showed" ➦ "Fig.7 shows";

  • Subchapter 2.7 (suggestions):

- "hepaic fibrogenic makers" ➦ "hepatic fibrogenic makers(?)/markers" (× 2);

  • Subchapter 3.3 (suggestions):

- "10 ml DMSO solution of IMB16-4 (30 mg/ml) was dropped" ➦ "10 ml DMSO solution of IMB16-4 (30 mg/ml) were dropped";

  • Subchapter 3.5 (suggestions):

- place the acronym explanation for BET (Brunauer–Emmett–Teller) in subchapter 2.2, as the first use of the term in this manuscript, not in subchapter 3.5;

  • Subchapter 3.8 (suggestions):

- "an atmosphere of 5% CO2" ➦ "an atmosphere of 5% CO2";

  • Subchapter 3.9 (suggestions):

- "MSNs were suspension in water" or "Another group of raw IMB16-4 was suspension in DMSO" ➦ "were suspended".

Author Response

I quite arrpeciate your favorite consideration. The revisions were addressed point by point below.

  • All of "in vivo" and "in vitro" entries have be written with italics.
  • In the text,  S.D. or SD (collation) has been replaced with  S.D.
  • Most of "anti-" in  the text have been replace with "resisting", "inhibitng", "agaist" and "for",seperately.
  • In the abstract,"because of the lack" has been replaced with "due to the lack".
  • In the abstract, "to have good potential of anti-liver fibrosis" has been  replaced with "to have good potential against liver fibrosis".
  • In the abstract, "These results provided powerful information for IMB16-4-MSNs as anti-liver fibrosis treatment in the furture"  has been replaced with "These results provided powerful information on IMB16-4-MSNs for the treatment of liver fibrosis in the future".
  • In introduction, "So for"has been replaced with "Therefore".
  • In introduction, "In this paper, we report for the first time the application of MSNs as the vehicle of IMB16-4 increased dissolution rate, improved bioavailability and anti-liver fibrosis effect"has been replaced with "In this paper, we report for the first time that MSNs were used as the carrier for IMB16-4 to increase the dissolution rate, improve the bioavailability and inhibit liver fibrosis effect".
  • In subchapter 2.3, "MSNs was soaked" has been changed to "MSNs were soaked". "The residua rate… were" has been  replaced with "The residual rates… were".
  • In subchapter 2.4, "These assume could be further confirmed by the results" has been replaced with  "These assumptions could be further confirmed by the results".
  • In subchapter 2.5, "may be mainly attributed to the large mesopores of MSNs" has been replaced with "may be mainly attributed to the large pores of MSNs".
  • In subchapter 2.6, "Fig.7 showed" has been replaced with "Fig.7 shows".
  • In subchapter 2.7, "hepaic fibrogenic makers"  has been  replaced with "hepatic fibrogenic markers" in the text.
  • In subchapter 3.3, "10 ml DMSO solution of IMB16-4 (30 mg/ml) was dropped" has been  replaced with "10 ml DMSO solution of IMB16-4 (30 mg/ml) were dropped".
  • In subchapter 3.5, we have placed the acronym explanation for BET (Brunauer–Emmett–Teller) in subchapter 2.2.
  • In subchapter 3.8, "an atmosphere of 5% CO2" has been  replaced with "an atmosphere of 5% CO2".
  • In subchapter 3.9, "MSNs were suspension in water" or "Another group of raw IMB16-4 was suspension in DMSO" has been replaced with  "were suspended".